# The Δ133p53 Isoforms, Tuners of the p53 Pathway

**DOI:** 10.3390/cancers12113422

**Published:** 2020-11-18

**Authors:** Sebastien M. Joruiz, Jessica A. Beck, Izumi Horikawa, Curtis C. Harris

**Affiliations:** Laboratory of Human Carcinogenesis, Center for Cancer Research, National Cancer Institute, National Institute of Health, Bethesda, MD 20892, USA; sebastien.jo@nih.gov (S.M.J.); jessica.beck@nih.gov (J.A.B.); horikawi@mail.nih.gov (I.H.)

**Keywords:** p53 isoforms, Δ133p53, p53, cancer, aging

## Abstract

**Simple Summary:**

*TP53*, the most frequently mutated gene in human cancers, has a key role in the maintenance of the genetic stability and, thus, in preventing tumor development. The p53-dependent responses were long thought to be solely driven by canonical p53α. However, it is now known that *TP53* physiologically expresses at least 12 p53 isoforms including Δ133p53α, Δ133p53β and Δ133p53γ. The Δ133p53 isoforms are potent modulators of the p53 pathway that regulate critical functions in cancer, physiological and premature aging, neurodegenerative diseases, immunity and inflammation, and tissue repair. This review aims to summarize the current knowledge on the Δ133p53 isoforms and how they contribute to multiple physiological and pathological mechanisms. Critically, further characterization of p53 isoforms may identify novel regulatory modes of p53 pathway functions that contribute to disease progression and facilitate the development of new therapeutic strategies.

**Abstract:**

The *TP53* gene is a critical tumor suppressor and key determinant of cell fate which regulates numerous cellular functions including DNA repair, cell cycle arrest, cellular senescence, apoptosis, autophagy and metabolism. In the last 15 years, the p53 pathway has grown in complexity through the discovery that *TP53* differentially expresses twelve p53 protein isoforms in human cells with both overlapping and unique biologic activities. Here, we summarize the current knowledge on the Δ133p53 isoforms (Δ133p53α, Δ133p53β and Δ133p53γ), which are evolutionary derived and found only in human and higher order primates. All three isoforms lack both of the transactivation domains and the beginning of the DNA-binding domain. Despite the absence of these canonical domains, the Δ133p53 isoforms maintain critical functions in cancer, physiological and premature aging, neurodegenerative diseases, immunity and inflammation, and tissue repair. The ability of the Δ133p53 isoforms to modulate the p53 pathway functions underscores the need to include these p53 isoforms in our understanding of how the p53 pathway contributes to multiple physiological and pathological mechanisms. Critically, further characterization of p53 isoforms may identify novel regulatory modes of p53 pathway functions that contribute to disease progression and facilitate the development of new therapeutic strategies.

## 1. Introduction

*TP53* gene is known as “the guardian of the genome” [1] highlighting its crucial role in maintaining genetic stability and preventing cancer formation. p53 is both a central sensor and a master regulator of cellular stresses. Its activation is adapted to the cell type and signal and induces a myriad of cellular responses such as DNA repair, cell cycle arrest, cellular senescence, apoptosis, autophagy and metabolism [2,3]. It is now known that wild-type *TP53* encodes, in tissue- and cell type-dependent manners, at least 12 different p53 protein isoforms: p53α (canonical p53), p53β, p53γ, Δ40p53α, Δ40p53β, Δ40p53γ, Δ133p53α, Δ133p53β, Δ133p53γ, Δ160p53α, Δ160p53β and Δ160p53γ [4].

The p53 family members interact with each other and can modulate each other’s expression and biological activities. This includes the p63 and p73 family proteins which can also interact to regulate several common target genes. For example, in the absence of p63 and p73 binding to p53-responsive promoters, p53 fails to induce apoptosis in response to DNA damage demonstrating the significance of p53 family member interactions in p53 pathway activity [5].

Although each family member is important to elicit the full spectrum of cellular responses regulated by the p53 pathway, we focus here on the regulation and activities of the Δ133p53 isoforms (Δ133p53α, Δ133p53β and Δ133p53γ). During the past decade of research in the p53 isoforms field, the Δ133p53 isoforms have been investigated in an array of contexts, including cancer and aging, and have emerged as key tuners of the p53 pathway in development and disease.

## 2. Regulatory Mechanisms of the Δ133p53 Isoforms

### 2.1. RNA Level

The mRNAs encoding the Δ133p53 isoforms are produced from the internal *TP53* promoter in the intron 4. Each of the Δ133p53 isoforms is translated from the methionine codon 133 and produced through alternative splicing of mRNA from exon-9 to either exon-10 (Δ133p53α), exon-9β (Δ133p53β) or exon-9γ (Δ133p53γ). This results in either the inclusion of the oligomerization and α C-terminal domains (62 amino-acids identical to those of canonical full-length p53α), the β C-terminal domain (10 amino acids), or the γ C-terminal domain (15 amino acids), respectively (Figure 1). In addition, mRNAs encoding the Δ133p53 isoforms can also be translated from the methionine codon 160, which produces each of the Δ160p53 isoforms with the same C-terminal variations.

The p53 isoforms are expressed in normal human tissues and are modulated by multiple mechanisms. Firstly, epigenetic events regulating the activity of the p53 proximal and internal promoters result in differential expression of Δ133/Δ160 α, β or γ mRNAs in normal human tissues [6]. In addition, the internal *TP53* promoter activity is influenced by polymorphisms, including the common pin3 (16-bp insertion in the intron-3) and R72P polymorphisms [7,8,9]. Furthermore, the internal promoter is transactivated by p53α and several p63/p73 family members and is repressed by p68, a co-activator of p53-dependent transcription, particularly following genotoxic stress [8,10,11]. Finally, splicing factors, such as SRSF1 and SRSF3, regulate the alternative splicing of *TP53* intron-9 by inhibiting retention of exon-9β/9γ [12,13]. This results in reduced production of the β and γ isoforms. Consistent with this function, treatment with TG003, a small drug inhibitor of the Cdc2-like kinases (Clk), prevents Clk-induced activation of SRSF1 and SRSF3 leading to increased exon-9β/9γ inclusion [13]. In contrast to SRSF1 and SRSF3, SRSF7 has been reported to enhance β splicing in response to ionizing radiation (IR) [14]. In this context, IR reduces SMG1 (a PI3K-like protein) binding on p53 pre-mRNA which enables the recruitment of RPL26. This allows recruitment of SRSF7 which favors the inclusion of exon 9β. Although these splicing factors primarily regulate the production of β/γ isoforms, SRSF1 has also been demonstrated to upregulate Δ133p53α expression without modulating canonical p53α [15], suggesting a possible link between the regulation of alternative transcription and the alternative splicing. However, the exact mechanism of this regulation is not known.

### 2.2. Protein Level

At the protein level, the differential inclusion of canonical p53 domains in each of the p53 isoforms affects their behavior. The Δ133p53 isoforms lack the first 132 amino-acids and are, therefore, devoid of the two transactivation domains (TADs) and part of the DNA-binding domain (DBD) (Figure 1). The DBD contains four conserved regions, the first of which spans between amino-acids 117 and 142, so the Δ133p53 isoforms lack most of this first region but retain the three others. These four conserved regions coordinate Zn^2+^ loops structures which are essential for p53 conformation [16,17]. This suggests that the Δ133p53 isoforms may have a different conformation as compared to their full-length counterpart. Although their conformation is not yet known, a recent study of the thermodynamic stability of Δ133p53β demonstrated that it has a higher aggregation propensity but can form a relatively stable complex with p53-specific DNA [18]. This suggests that there may be some differences in Δ133p53β conformation promoting aggregation but not enough to inhibit its binding to p53 response elements. The C-terminal region contains the hinge domain (HD), the oligomerization domain (OD), and the α regulatory domain (Figure 1). In the β and γ isoforms, part of the OD and the α domain are replaced by the β or γ domains, respectively. The lack of the OD in the β and γ isoforms may impair their ability to oligomerize with other isoforms. However, p53β and p53γ can still form DNA-mediated complexes with p53α in the presence of p53-responsive promoter suggesting that the ability of these isoforms to interact with p53α is not completely abolished [13]. In addition, the differences between the α, β or γ domains may impact cellular localization of the isoforms. In p53α, there are three nuclear localization sequences (NLS): one in the HD and two in the α-domain (Figure 1). The Δ133p53α protein, which also contains all three NLS, is primarily located in the nucleus [6]. Although the Δ133p53β lacks two of the three NLS, it contains the HD-associated NLS and is detected in the cytoplasm and in the nucleus where it forms speckles. Interestingly, Δ133p53γ has the same NLS as Δ133p53β; however, it is mostly detected in the cytoplasm. This indicates that β and γ domains are not equivalent and may suggest that either the β peptide modifies the subcellular localization or that the γ peptide counteracts the NLS activity.

Although MDM-2 binds to p53 mainly in the N-terminal TAD, it also contains a secondary binding site in the DBD which is common to all isoforms [19]. Camus et al. confirmed that MDM-2 binds to all p53 isoforms, including the Δ133p53, without promoting their degradation [20]. Furthermore, they reported that all p53 isoforms are ubiquitylated and degraded by the proteasome with different kinetics in a cancer cell line. Hence, Δ133p53γ has the shortest half-life of all the isoforms, about 40 min, while Δ133p53α and Δ133p53β have half-lives of 110 and 135 min, respectively. The half-lives of all three Δ133p53 isoforms were increased upon inhibition of the proteasome by MG-132 treatment, although the increase was smaller for Δ133p53β. It is, therefore, expected that MDM-2 can also block the transcriptional activities of most isoforms either directly or by recruiting corepressors such as hCtBP2 [21,22]. In addition to proteasome degradation, Δ133p53α is primarily degraded by chaperone-mediated autophagy during replicative senescence in normal human cells including fibroblasts, astrocytes and T-cells [23,24,25]. This process may be inhibited by direct interaction of the E3 ubiquitin ligase STUB1/CHIP with Δ133p53α or by pharmacological inhibition of autophagy by bafilomycin A1 [23].

Finally, one of the major regulatory systems of p53 stability and activity is post-translational modifications (PTM). The PTMs occurring on p53α and their implications for its stability and activity have been extensively studied and described [26,27,28]. However, it is not known whether the same modifications occur on Δ133p53 isoforms and, if so, whether they have similar effects. Given that the Δ133p53 isoforms retain several of the PTM sites described in p53α, and since the β (DQTSFQKENC) and γ (MLLDLRWCYFLINSS) C-terminal sequences also contain several serines, threonines, lysines or arginines residues, it is expected that several PTM can occur on all isoforms. In support of this expectation, p53β and p53γ appear as several dots with different isoelectric points when visualized on 2D gels, suggesting different charges due to post-translational modifications [29,30,31,32]. In addition, as mentioned earlier, Camus et al. have reported that all p53 isoforms are ubiquitylated without necessarily promoting their degradation [20]. This suggests that ubiquitylation of p53 isoforms may also be associated with proteasome-independent functions, including the regulation of subcellular location and protein interaction [33,34], further underscoring the potential for PTMs to modulate the stability and activity of p53 isoforms.

## 3. The Δ133p53 Isoforms in Cancers

Clinical sample studies widely rely on qRT-PCR methods to determine the expression of mRNAs. Therefore, it is important to consider the technical limitations of the current detection methods. The first limitation is the difficulty to detect p53 isoforms expressed at lower abundance in patient samples or in RNA-seq databases [35]. Secondly, to specifically detect and quantify each of Δ133p53α, Δ133p53β and Δ133p53γ mRNAs separately, a simultaneous discrimination of N-terminal events (i.e., the transcriptional initiation for Δ133p53/Δ160p53 versus that for p53α and Δ40p53) and C-terminal events (i.e., α, β or γ splicing) is required. Since Δ133p53α, Δ133p53β and Δ133p53γ mRNAs share a common region spanning 618 base-pairs, such simultaneous discrimination by qRT-PCR would need to amplify long amplicons, which may be difficult to obtain with conventional PCR methods and clinical RNA samples that could be partially degraded. Therefore, earlier clinical studies have only quantitated total expression levels of all three Δ133p53 mRNAs without distinguishing between α, β or γ transcripts (Table 1, upper). Recent technical developments, such as RNAscope (RNA in situ hybridization), chromatography-tandem mass spectrometry and multiplex long amplicon droplet digital PCR, are beginning to circumvent these limitations [36,37,38] (Table 1, lower).

Given their potency to regulate crucial aspects of the p53 response, investigation of associations between the Δ133p53 isoforms and cancer patient prognosis started shortly after their identification (Table 1). In colorectal cancers, total Δ133p53 mRNAs are downregulated in colon adenomas compared to non-tumor cells, but their expression is upregulated during the progression from adenoma to carcinoma [39]. This suggests that the total Δ133p53 isoforms may not promote benign tumor formation but may play a role in progression from benign to malignant tumors. Consistently, increased expression of total Δ133p53 mRNAs is associated with shorter disease-free survival and correlate with higher risks of recurrence [40,41]. This could be due to Δ133p53α-mediated activation of JAK-STAT3 and RhoA-ROCK signaling promoting colorectal cancer cell growth and invasion [41]. In addition, Δ133p53β protein binds to and inhibits RhoB, thus preventing RhoB-induced apoptosis [40]. This may have important implications in tumor response to therapies such as camptothecin [40]. This is further supported by data from analysis of cholangiocarcinomas in which shortened overall patient survival is correlated with upregulation of total Δ133p53 mRNAs and downregulation of full length p53 mRNAs (p53α/p53β/p53γ) [42]. Interestingly, targeting the Δ133p53 isoforms restores sensitivity to 5-Fluorouracil in resistant cholangiocarcinomas [43]. A possible explanation for this is the ability of Δ133p53α to enhance DNA repair [52,53,54], which counteracts the effect of 5-FU. Critically, Δ133p53α may have a similar impact on other DNA-damage inducing cancer therapies. Similar upregulation of total Δ133p53 mRNAs has been observed in B-cell precursor acute lymphoblastic leukemia [55] and in lung carcinomas as compared to adjacent non-cancerous tissue [44] and is reported to increase cancer cell survival in pathogen-driven cancers such as gastric tumors associated with *H. pylori* [56]. In the case of gastric cancer, total Δ133p53 mRNAs expression correlates with the NF-κB p65 subunit expression and may promote gastric cell growth [57]. Recently, a high total Δ133p53/TAp53 ratio was shown to be a marker of poor overall and progression-free survival in esophageal squamous cell carcinoma [45]. Furthermore, specific changes in this ratio were correlated to clinical events. For example, the ratio quickly diminished following surgical resection of the tumor and increased upon tumor recurrence. Interestingly, this ratio was measured as circulating RNAs in patient’s serum, which would represent a non-invasive, real-time method to monitor patient’s recurrence.

Importantly, total Δ133p53 mRNAs expression is not always a bad prognostic marker. In mutant *TP53* serous ovarian cancer, patients expressing total Δ133p53 mRNAs have longer overall survival and disease-free survival [46,58]. Interestingly, a recent study on high-grade serous ovarian cancer established that high expression of total Δ133p53 mRNAs is correlated with increased overall survival independently of *TP53* mutation status [47]. This suggests that the association between total Δ133p53 mRNAs expression and good prognosis in serous ovarian cancers does not depend exclusively on *TP53* mutation status. This theory is reinforced by another recent publication which reported that wild-type (WT) *TP53* renal cell carcinomas (RCC) have a worse overall survival than those that harbor p53 mutations [48]. Interestingly, total Δ133p53 mRNAs were found downregulated in WT tumors as compared to non-tumor adjacent tissue, while they were unaffected in mutant tumors. This suggests that the worse prognosis in WT RCC may be due to the reduction of WT Δ133p53 expression.

In all the above studies, the clinical associations have been made with the expression level of all Δ133p53 mRNAs. Therefore, these associations cannot be linked to a biological activity of a specific isoform without cell-based functional assays (described below). Nevertheless, some studies using recent techniques of RNA quantification gathered insights into a possible role of individual Δ133p53 isoforms (Table 1, lower). For example, using nested-PCR, Δ133p53α mRNA is detected in breast cancer samples but not in normal breast tissue [6]. However, it has also been shown that Δ133p53α is neither oncogenic nor mutagenic in normal human cells (Table 2) [54], which indicates that it may be involved in cancer progression once developed, but not in cancer formation. In breast cancer patients, detection of Δ133p53β mRNA by nested-PCR is associated with worse overall and disease-free survival [49]. This was explained by its propensity to promote cancer invasion by triggering epithelial to mesenchymal transition [49,50]. Using the new RNAscope in situ hybridization and chromatography-tandem mass spectrometry techniques, increased Δ133p53β expression was found to stimulate glioblastoma and prostate cancer development and aggressiveness by promoting an immunosuppressive and chemo-resistant microenvironment [36,37]. In prostate cancer, increased Δ133p53β mRNA expression is associated with shorter progression-free survival [37]. Its increased expression has also been associated with poorer overall survival in melanoma [51].

## 4. Biological Activities

Since the discovery of human p53 isoforms, several studies have analyzed the activity of Δ133p53α, Δ133p53β, and to a much lesser extent, Δ133p53γ. These isoforms have emerged as important players of the p53 pathway and are involved in key processes including DNA repair, senescence, stemness, aging and neurodegeneration (Table 3).

### 4.1. Δ133p53α

#### 4.1.1. Δ133p53α in Cellular Senescence and SASP

Δ133p53α is primarily recognized as a regulator of cellular senescence. Senescence is characterized by increased senescence-associated β-galactosidase activity, increased expression of cell-cycle arrest factors (e.g., miR-34, p16 and p21), and adoption of the senescence-associated secretory phenotype (SASP) [69]. Senescent cells harbor a p53 isoforms signature characterized by increased p53β and decreased Δ133p53α proteins [23,24,25,39,61,62]. In this context, p53β cooperates with p53α to promote the induction of cellular senescence while Δ133p53α acts as a dominant-negative inhibitor of p53α-mediated senescence [23,24,25,39,61,62]. Interestingly, in all these articles, Δ133p53α overexpression leads to reduction or even reversion of senescence, demonstrated by reduced secretion of SASP factors, restoration of homeostatic functions, and extension of the replicative lifespan. This was verified in a variety of normal human cells, including astrocytes, fibroblasts and T-cells. This activity could be particularly important in neurodegenerative and premature aging diseases in which cellular senescence contributes to disease progression, including multiple sclerosis and Alzheimer’s disease [59,70,71]. Specifically, loss of Δ133p53α and increased expression of p53β has been observed in the brain tissue of Alzheimer’s disease and amyotrophic lateral sclerosis patients as compared to age-matched normal brain tissue [25]. In this context, astrocyte senescence has been suggested to induce neurotoxicity while restoration of ∆133p53α increases the production of neurotrophic factors, such as NGF or IGF-1, and reduces SASP protein expression, including proinflammatory cytokines such as IL-6, to promote neuroprotection [25]. This could have critical implications in cancer patients receiving cranial radiotherapy where radiation-induced DNA damage promotes astrocyte senescence [61]. Furthermore, it was recently reported that, in the absence of cellular stress, Δ133p53α represses the transcription of senescence-associated p53α target genes but not the expression of apoptosis genes [54]. Importantly, this is consistent with the hypothesis that the dominant-negative effect of Δ133p53α towards p53α may be senescence-specific. Furthermore, if the stress is mild, Δ133p53α may not antagonize full-length p53 activity, but coordinate with it to promote cell survival by inducing antioxidant gene expression [63].

#### 4.1.2. Δ133p53α in DNA Repair

Δ133p53α is also involved in DNA repair, particularly of DNA double strand breaks (DSB). Following genotoxic stress, Δ133p53α accumulates [8,11] and leads to upregulation of repair genes, such as LIG4, RAD51, and RAD52 and the activation of DNA repair pathways [52,60]. This has been suggested to represent a p53α-independent function of Δ133p53α that occurs through interaction with the p53 family member p73 [60]. However, canonical p53α represses the DSB repair pathways [72,73] and promotes senescence and apoptosis following DSB damage [2,3] suggesting that the DNA repair activity of Δ133p53α may also occur through p53α-dependent mechanisms. It was shown that Δ133p53α interacts with full-length p53α and represses p53α-mediated apoptosis and senescence and favors DNA-repair and cell cycle progression [6,39]. Furthermore, a recent report demonstrated that ∆133p53α inhibits 5-Lipooxigenase, leading to enhanced DNA repair and neuroprotection [67]. Thus, 5-lipooxygenase is involved in Alzheimer’s disease development, including Aβ amyloid deposition and tau hyper-phosphorylation, leading to oxidative stress and DNA damage, a key aspect of the disease development [74,75]. Therefore, enhancing DNA repair is another mechanism by which Δ133p53α may protect from the development of neurodegenerative disease.

#### 4.1.3. Δ133p53α in Pluripotent Stem Cells Regulation

Human induced pluripotent stem-cell (iPSC) and embryonic stem-cell (ESC) lines have higher Δ133p53α expression than the normal fibroblasts from which the iPSC were derived [54]. Following transduction of the Yamanaka factors (Oct4, Klf4, c-Myc, and Sox2), fibroblasts upregulate Δ133p53α during reprogramming, suggesting that Δ133p53α is a significant and an early player in this process. Consistently, overexpression of Δ133p53α with the Yamanaka factors increases reprogramming efficiency and decreases chromosomal aberrations, as compared to Yamanaka factors alone [53,54]. In contrast, Δ133p53α knockdown resulted in decreased reprogramming and increased genomic instability. This was attributed to the ability of Δ133p53α to inhibit senescence-associated p53α-target genes (p21, PAI-1, IGFBP7, miR-34a) and to promote DNA repair which allows cells to keep replicating while ensuring genomic stability. Δ133p53α overexpression also leads to fewer chromosomal aberrations and somatic mutations than full-length p53 knockdown [54]. This suggests that expression of Δ133p53α represents a non-oncogenic endogenous mechanism to obtain normal pluripotent stem cells which could be used in regenerative medicine to treat age-associated tissue dysfunction or degenerative diseases, such as sarcopenia and muscular dystrophy. Consistent with this hypothesis, SRSF1-mediated Δ133p53α overexpression promotes neointima formation after vascular injury by interacting with EGR1 (early-growth-response gene 1) to activate KLF5 (Krüppel-like factor 5), leading to vascular smooth muscle cell proliferation [15].

#### 4.1.4. Δ133p53α in Cancer Biology

In the context of cancer, Δ133p53α and Δ133p53γ, but not Δ133p53β, can promote angiogenesis and subsequent metastasis by up-regulating expression of pro-angiogenic genes and repressing anti-angiogenic genes [66]. This is another critical function of Δ133p53α which has been suggested to occur independently of p53α although the exact mechanism is unknown. Since the Δ133p53 isoforms lack transactivation domains, a possible explanation for this activity could be though interaction with p53 family members other than p53α. For example, it was shown that all three Δ133p53 isoforms can interact with TAp73α and TAp73β and counteract TAp73β transactivation function in a promoter-dependent manner [76]. Interestingly, Δ133p53α and Δ133p53β inhibits TAp73β transactivation of pro-apoptotic genes [76]. In addition, it was shown that WT Δ133p53α overexpression increases migration of p53-null human osteosarcoma cells [68]. Furthermore, this was validated in WT colorectal cancer cells where individual overexpression of either Δ133p53α, Δ133p53β or Δ133p53γ increased cell invasion and that this required RhoA-ROCK activity [41]. Interestingly, among the up-regulated genes in the study by Bernard et al. [66] are secreted factors HGF, angiogenin and VEGF, which are pro-angiogenic genes, but also common components of SASP. The upregulation of these factors by Δ133p53α is contrary to the commonly described SASP-inhibiting functions of Δ133p53α and may suggest that the regulation of some secreted factors by Δ133p53α can vary in a context-dependent manner (e.g., angiogenesis or SASP). Altogether these findings suggest that Δ133p53α may function differently in the context of cancer.

### 4.2. Δ133p53β

In contrast to the role of Δ133p53α in normal pluripotent stem cells (described above), Δ133p53β is associated with an enhanced cancer stem cell phenotype. In breast cancer cells, Δ133p53β stimulates the expression of the key pluripotency factors SOX2, OCT3/4, and NANOG leading to mammosphere formation and increased metastatic capacity [65]. Upon treatment with etoposide, increased Δ133p53β expression is associated with upregulation of the pluripotency genes which may increase the risk of treatment resistance and tumor recurrence. While promoting cancer stemness, Δ133p53β also favors cellular dysdifferentiation by inducing epithelial to mesenchymal transition (EMT) in cancer cells [49], which involves genome-wide epigenetic reprogramming [77,78]. The expression of Δ133p53β is increased in highly invasive breast and colon cancer cells, regardless of *TP53* mutation status, while its loss reduces tumor cell invasion. Furthermore, depletion of Δ133p53β leads to a significant induction of E-cadherin and a decrease in vimentin expression. This suggests a global reversion of EMT-cell reprogramming upon loss of Δ133p53β. Therefore, elevated expression of Δ133p53β may represent a critical determinant of cancer invasiveness and recurrence, and poor cancer survival in breast cancer patients [49]. This pro-tumor role for Δ133p53β has also been corroborated by studies in glioblastoma and prostate cancer [36,37].

The results described herein demonstrate that Δ133p53α and Δ133p53β share some common functions including the promotion of cellular proliferation and motility [15,24,39,41,49,62,64,66,68] and the regulation of stem cell potential [53,54,65], although they are likely to be cell type- and context-dependent. However, some functions have not been attributed to both isoforms. For example, Δ133p53α is involved in the regulation of cellular senescence [24,39,62,64] and protects against DNA damage [52,54,60] while similar functions have not yet been described for Δ133p53β. In addition, Δ133p53β is reported to promote aggressive cancer phenotypes by favoring an immunosuppressive environment [37] while comparable functions have not been attributed to Δ133p53α. The specific DNA-binding ability of Δ133p53α [52] and Δ133p53β [18] may contribute to some of the common and unique functions. Although much less is known about Δ133p53γ, several experiments reported that Δ133p53γ may also regulate cell proliferation and motility and promote angiogenesis to a lesser extent than Δ133p53α or Δ133p53β [41,49,66]. There is, therefore, a critical need for further research on the common and isoform-unique functions of Δ133p53α, Δ133p53β and Δ133p53γ and the underlying mechanisms.

## 5. N-Terminally Truncated p53 Isoforms through Evolution: Models for Human Δ133p53 Isoforms?

The dual gene structure of the *TP53* gene with the proximal and internal promoters is highly conserved through evolution [6]. The internal promoter is found in *TP63* and *TP73* genes which have evolved with *TP53* from a common ancestral gene [79]. It is also found in different species, such as *Drosophila melanogaster* and Zebrafish [80,81]. Interestingly, as a result of molecular evolution, Δ133p53 isoforms are specifically present in humans and high order primates, but not in other organisms including laboratory animals (e.g., mice and rats) and long-lived animals (e.g., elephants, which have increased copy numbers of *TP53* [82]). This is due to the lack of an initiating methionine codon corresponding to the human codon 133, which is mostly a leucine in other organisms (Figure 2A). Unlike the methionine 133, the methionine 160 is conserved through all species in Figure 2A except for African clawed frog and Zebrafish, which are the farthest from primates in evolution. This suggests that the production of Δ133p53 isoforms may be a more recent evolutionary event. Several animal models which have been considered for the study of N-terminally truncated p53 isoforms include drosophila, mouse and zebrafish.

### 5.1. Drosophila

*Drosophila melanogaster* has p53, which is distantly related in sequence but highly homologous in overall structure to human p53, and expresses an N-terminally truncated isoform, ΔNp53 (Figure 2B). In a fashion reminiscent of human Δ133p53 isoforms, *Drosophila* ΔNp53 lacks the first 122 amino-acids and is expressed from the internal promoter. However, in contrast to human Δ133p53, the initiating methionine and the first 13 amino acids of ΔNp53 are encoded by a cryptic exon and therefore do not appear in the full-length protein. Furthermore, because ΔNp53 starts in the transactivation domain, it contains the entire DNA binding domain. Therefore, ΔNp53 corresponds better to human Δ40p53 isoforms than Δ133p53.

### 5.2. Mouse

Mouse p53 contains a methionine at codon 120. However, the Δ120p53 isoform has not been observed to be expressed endogenously. To circumvent this issue, Slatter et al. thus performed genetic engineering of exons 3 and 4 of the mouse *Trp53* gene to obtain mice expressing an N-terminally truncated version of p53 (named Δ122p53) that mimics this putative isoform [83]. The studies using these engineered mice have shown that Δ122p53 promotes tumor invasion and metastasis by regulating secretion of pro-inflammatory IL-6 and CCL2 [68] and that Δ122p53-induced IL-6 secretion activates JAK-STAT3 and RhoA-ROCK signaling leading to higher tumor incidence and metastasis [41]. Of note, these tumorigenic activities of Δ122p53 in mice are in marked contrast to the non-oncogenic nature of human Δ133p53α in normal human cells (Table 2). Several experimental and biological factors may possibly be involved in these apparently different activities of Δ133p53α and Δ122p53. While Δ133p53α was examined mainly in the presence of full-length p53α, some of Δ122p53 data were from Δ122p53 homozygous mice without full-length p53α [41,68,83], therefore overlooking p53α-dependent functions of Δ122p53. In addition, there may be cell type- and context-dependent functional differences or species-specific differences that influence their activities. Finally, it is possible that Δ133p53α and Δ122p53 intrinsically differ in some functions due to their different N-terminal ends and different transcriptional controls.

The human p53 knock-in (Hupki) mice [84] are a possible mouse model where in vivo functions of human Δ133p53 isoforms may be examined under more physiological settings. In the Hupki mice, the mouse exons 4 to 9 were replaced by the homologous human p53 sequence leading to the production of a chimeric canonical p53α protein, allowing for examination of p53α-dependent functions [84]. Importantly, the inserted human sequence contains the *TP53* internal promoter (in human intron 4) and methionine codons 133 and 160 (in human exon 5) but lacks the *TP53* exons 9β and 9γ. As such, these mice potentially express chimericΔ133p53α- and Δ160p53α-like isoforms containing the human core domain and the mouse C-terminal regions. The expression analysis of these chimeric isoforms in the Hupki mice and their phenotypic examination (e.g., aging and tumor incidence), in parallel to the generation of isoform-specific transgenic mice, are particularly important to establish in vivo models for further in-depth studies of Δ133p53 isoforms.

### 5.3. Zebrafish

Zebrafish expresses Δ113p53 (Figure 2A,B), which is produced from the internal promoter located in the intron 4, misses the transactivation domain and part of the DNA binding domain, and is transactivated by the full-length p53 protein [80,85]. These make zebrafish Δ113p53 similar to human Δ133p53. Like its human counterpart, Δ113p53 leads to upregulation of DNA repair genes, including LIG4, RAD51 and RAD52, and the activation of DNA repair pathways following genotoxic stress [52]. In order to favor DNA DSB repair, Δ113p53 interacts with full-length p53 to re-orientate its activity towards DNA repair by differentially modulating its target genes expression [86]. This way, Δ113p53 can shift p53 binding to the anti-apoptotic BCL2 promoter and induce expression of cell cycle arrest proteins (i.e., cyclin-G1 and p21) while repressing BAX or Reprimo pro-apoptotic genes [87]. Induction of cellular senescence and differentiation maintains critical roles in embryogenesis. Zebrafish Δ113p53 and Human Δ133p53α are both repressing cellular senescence and favoring normal cells pluripotency. As such, Δ113p53 overexpression prevents cellular differentiation and thereby induce embryogenesis defects [88]. Furthermore, in zebrafish, Def gene mutations selectively up-regulates Δ113p53 expression, leading to impaired development of digestive organs [80]. A similar mechanism has been described with mutation of DHX15, a splicing factor, which up-regulates Δ113p53 expression and leads to morphological defects and embryo lethality [89]. On the contrary, in differentiated tissues, homeostasis is maintained by the pluripotency and differentiation potential of resident cells, which is favored by Δ133p53α/Δ113p53. Importantly, a recent report demonstrated that Δ113p53 promotes zebrafish heart regeneration [90]. It accumulates in cardiomyocytes at the injury site, favoring their proliferation and the maintenance of redox homeostasis, thus promoting myocardial regeneration. While zebrafish Δ113p53 currently provides the only one vertebrate model of a naturally expressed Δ133p53-like isoform, the development of mammalian models is critically needed to study human Δ133p53 isoforms in evolutionarily and biologically closer species.

## 6. Conclusions and Perspectives

The tumor suppressor role of the p53 pathway is underscored by the numerous functions it regulates, including DNA repair, cell cycle arrest, cellular senescence, apoptosis, autophagy and metabolism. The discovery that *TP53* differentially expresses twelve different p53 protein isoforms and the study of their biological activities have provided critical insights to our understanding of the p53 pathway and, importantly, have further advanced the role of p53 beyond cancer. Naturally, the temptation to categorize each isoform as “good” or “bad” quickly arose. Most studies about Δ133p53 isoforms have been done in the context of cancer where they are frequently linked to bad prognosis. In addition, they can exert dominant-negative activity towards a subset of p53α functions. Therefore, uncontrolled activities of the Δ133p53 isoforms are generally considered detrimental and tumor-promoting. However, it is inappropriate to define a p53 isoform simply as an oncogene or tumor suppressor since its activity depends on the cell context [91]. This is further complicated as the function of p53 isoforms varies in different conditions (e.g., physiological conditions, cancer microenvironments or neurodegenerative diseases) such that a specific function of a given isoform may be beneficial in one context and detrimental in another. Furthermore, p53α has multiple protein modifications that modulates its activity [26,27,28] and could similarly affect the Δ133p53 isoforms. Future studies should investigate if such protein modifications occur and alter functions of the Δ133p53 isoforms.

Here, we propose a model in which, under physiological conditions, the Δ133p53 isoforms are involved in the maintenance of cell and tissue homeostasis (Figure 3). Δ133p53α protects tissues from degeneration and aging-associated changes by preventing the cells from undergoing senescence and SASP-induced inflammation while favoring DNA repair of damaged cells [39,52,54]. An example of this is the maintenance of astrocytes in a replicative, neuroprotective state which prevents neurodegeneration [25,62]. Furthermore, Δ133p53α does not inhibit p53α-dependent apoptosis of severely damaged cells and, importantly, Δ133p53α ensure the maintenance of pluripotent stem cell reserves to promote tissue integrity [44,56]. As previously mentioned, Δ133p53α is neither oncogenic nor mutagenic [54], and its overexpression does not cause immortalization or malignant transformation (Table 2). Under physiological conditions, it may even protect against the development of cancer. This may initially seem paradoxical since senescence is considered a major barrier to tumorigenesis by inhibiting the replication of aged and DNA-damaged cells [92]. However, Δ133p53α maintains genomic integrity by promoting DNA repair activity while allowing p53α-mediated apoptotic activity to remove severely damaged cells which cannot be appropriately repaired. In addition, when senescent cells accumulate and are not efficiently removed, persistent SASP secretion induces chronic inflammation, tissue fibrosis, angiogenesis, and cellular proliferation while reducing viability of neighboring cells [69,93]. Critically, all these functions favor cancer development. Therefore, by inhibiting senescence and SASP in normal cells, Δ133p53α prevents the establishment of a cancer prone microenvironment. Lastly, ∆133p53α expression in CD8+ T-cells protects them from T-cell exhaustion and senescence [24], possibly facilitating the detection and clearance of cancer cells and enhancing efficiency of cancer immunotherapy [94,95]. Each of these mechanisms underscores the critical anti-tumor functions of Δ133p53α.

However, once a tumor is developed, the Δ133p53 isoforms may promote tumor progression through the same activities (Figure 3). For example, Δ133p53α prevents the cancer cells from entering senescence which is a barrier to tumor development and malignant progression [39,92,96,97] and which could be used as an effective cancer therapeutic option [98]. Furthermore, most non-targeted chemotherapeutic drugs rely on genotoxic effects that may be counteracted by the DNA repair mechanisms and the cancer stemness phenotype, which are promoted by Δ133p53α and Δ133p53β, respectively [43,49,65]. This may promote tumor malignancy and resistance to treatment. In addition, Δ133p53β also promotes immunosuppressive microenvironment [36,37] and chronic inflammation [41,68,99,100], both of which favors malignant progression. All three Δ133p53 isoforms may also facilitate cancer metastasis by upregulating the secretion of angiogenic factors and vascular smooth muscle cells proliferation, promoting cancer cell EMT, and increasing cancer cell motility and invasion potential [15,41,49,65,66,68].

These examples underscore the complexity of the p53 isoform story. As such, many questions remain. For example, how can Δ133p53α repress the secretion of SASP factors in one context [23,24,25,39,54,61,62] while promoting the secretion of IL-6 and pro-angiogenic factors in another [41,66,68,99,100]? Theoretically, these differences may be explained by the SASP secretome which varies based on the senescent cell type, tissue of origin and the inducer of senescence (e.g., oncogene activation, accumulated DNA-damage, telomeres shortening and oxidative stress) [101]; however, the molecular mechanisms behind these differences remain unknown.

Furthermore, several p53 isoforms are generally co-expressed in human tissue [6]. Hence, the activities of Δ133p53 isoforms may be modulated or even counterbalanced by the activities of co-expressed isoforms. For example, while Δ133p53α prevents cellular senescence, p53β promotes it [24,25,39,61,62]. Whether a cell undergoes senescence or not likely depends on the balance between the two isoforms. Similarly, Δ40p53α expression may counteract Δ133p53α activities. For example, Δ40p53α promotes senescence and neurodegeneration by increasing tau phosphorylation or promoting the production of amyloid-beta [102,103,104]. Expression of Δ40p53α is also associated with premature aging and a shorter lifespan through abnormal IGF signaling and increased senescence [105,106]. This underscores the importance of not only analyzing the activities of each individual isoform but also the interplay between co-expressed isoforms to fully understand the functions of the p53 pathway in specific disease contexts.

Finally, according to the IARC (International Agency for Research on Cancer) somatic mutations database [107], about 92% of *TP53* single point mutations in human cancers are between codons 133 and 331 and, therefore, affect most p53 isoforms. As such, exploring the effect of mutations on Δ133p53 isoforms activities is a critical, albeit unexplored area in the p53 field. The importance of characterizing mutant p53 isoforms is further supported by several clinical studies demonstrating that the accuracy of prognosis for cancer patients could be greatly improved by combining *TP53* mutation status and p53 isoforms expression.

Due to their potent and wide-ranging activities, the Δ133p53 isoforms have been identified as major tuners of the p53 pathway. Better understanding of their molecular mechanisms could be the key to improving our understanding of the p53 pathway and to validating each of the Δ133p53 isoforms as therapeutic targets in cancer or degenerative diseases.

## Figures and Tables

**Figure 1 cancers-12-03422-f001:**
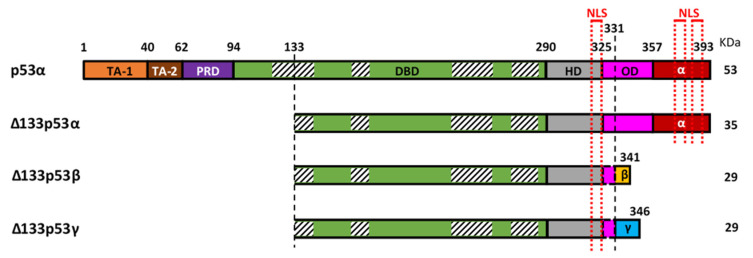
Representation of canonical p53 (p53α) and the three Δ133p53 isoforms (Δ133p53α, Δ133p53β and Δ133p53γ). p53α contains two transactivation domains (TAD1 and TAD2), the proline-rich domain (PRD), the DNA binding domain (DBD), the hinge domain (HD), the oligomerization domain (OD) and the α-domain. Part of the OD and the α-domain can be alternatively spliced to create the β and γ isoforms. The Δ133p53 isoforms lack TA-1, TA-2, PRD and part of the DBD. The DBD contains 4 highly conserved regions (striped boxes) spanning amino-acids 117–142, 171–181, 234–258 and 270–286 respectively, and the Δ133p53 isoforms lack, therefore, the majority of the first conserved region. Three nuclear localization sequences (NLS) are in the C-terminal part (red dotted lines), one in the HD and two in the α-domain, so the β and γ isoforms only have the first one.

**Figure 2 cancers-12-03422-f002:**
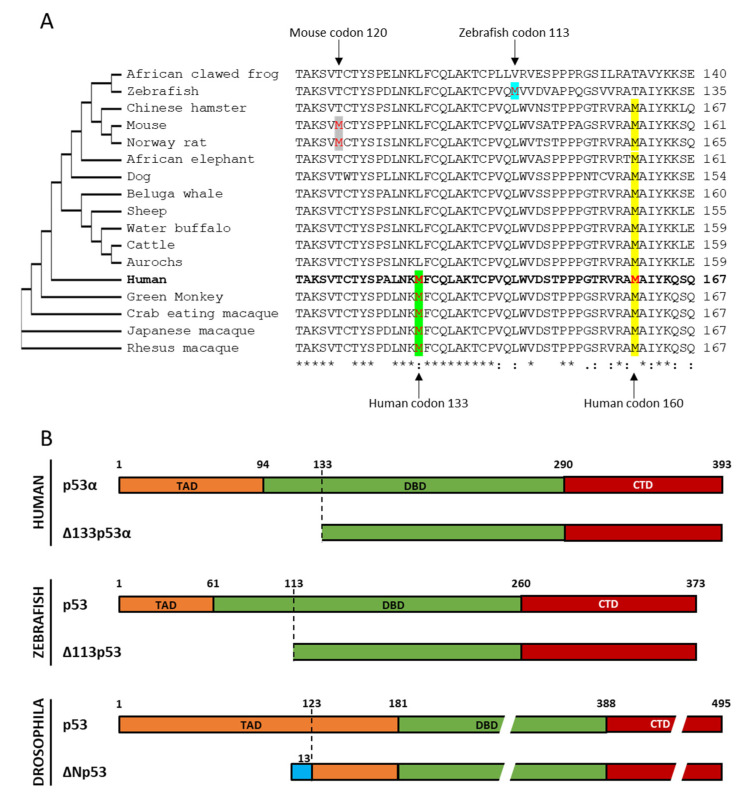
Δ133p53 isoforms through evolution. (**A**) Alignment of human p53 sequence (accession number NP_001119584) between amino-acid 118 and 167 with the corresponding sequence in African clawed frog (CAA54672), Zebrafish (NP_571402), Chinese hamster (AAC53040), Mouse (BAA82344), Rat (NP_112251), African elephant (XP_010594888.1), dog (BAA78379), Beluga whale (AAL83290), Sheep (CAA57349), Water buffalo (AEG21062), Cattle (CAA57348), Aurochs (BAA08629), Green monkey (XP_008008385), Crab eating macaque (AAB91535), Japanese macaque (AAN64028) and Rhesus macaque (XP_005582844). The phylogenetic tree on the left is a neighbor joining tree without distance correction. The phylogenetic tree and the alignment were designed using Clustal Omega multiple sequence alignment online tool (https://www.ebi.ac.uk/Tools/msa/). (**B**) To scale representation of N-terminally truncated isoforms in human, zebrafish and drosophila as compared to their respective full-length proteins. Proteins are divided in 3 functional domains: transactivation domain (TAD) in orange, DNA binding domain (DBD) in green and C-terminal domain (CTD) in red. Of note, the identical colors in three species highlight the common functional domains with varying degrees of sequence homology.

**Figure 3 cancers-12-03422-f003:**
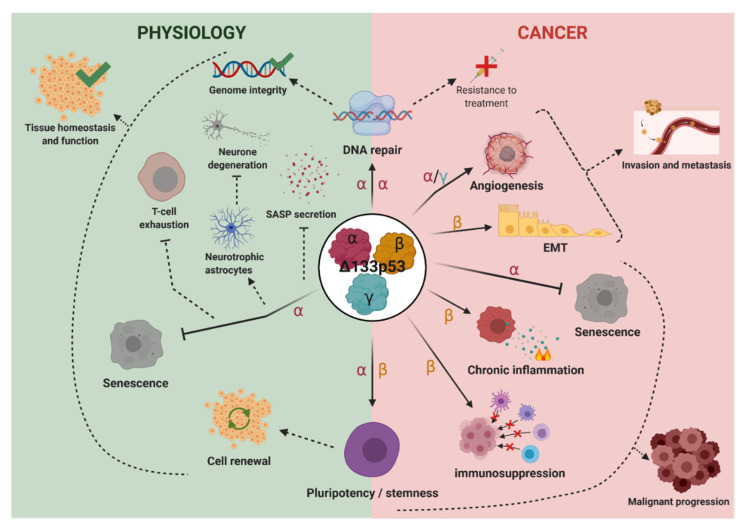
Proposed model for the functions of the three Δ133p53 isoforms. Under physiological conditions, Δ133p53α prevents replicative senescence while ensuring genomic integrity of the replicative normal cells. By reducing the number of senescent cells, it reduces senescence-associated secretory phenotype (SASP) secretion which prevents the development of cancer prone microenvironment. Δ133p53α also maintains the astrocytes in a replicative, neuroprotective state which prevents neurodegeneration. Furthermore, Δ133p53α protects CD8+ T-cells from T-cell exhaustion and senescence thereby facilitating cellular immunity against infectious diseases and cancer. Lastly, Δ133p53α isoforms may ensure the maintenance of pluripotent stem cell reserves which is critical for cell renewal in tissues. Taken together, all these activities of Δ133p53α under physiological conditions contribute to normal tissue homeostasis and functions. In the context of cancer, Δ133p53 isoforms may promote tumor progression and aggressiveness. By promoting DNA repair, Δ133p53α isoform may counteract the effect of genotoxic drugs leading to treatment resistance. All three Δ133p53 isoforms promote cancer cell invasion and metastasis by favoring either angiogenesis, vascular smooth muscle cell proliferation and/or epithelial to mesenchymal transition. Furthermore, Δ133p53β promotes cancer stemness phenotype, immunosuppressive microenvironment and chronic inflammation, while Δ133p53α may prevent cancer cells to enter senescence, thus favoring malignant progression.

**Table 1 cancers-12-03422-t001:** Expression of Δ133p53 isoforms in human cancer associated with disease progression, therapy response and patient prognosis.

Cancer Type	Total Δ133p53 mRNAs (α, β and γ) Association with Disease	Ref.
Colon cancer	Total Δ133p53 mRNAs are downregulated in adenoma vs. normal tissue, but overexpressed in carcinoma vs. adenoma or normal tissue	[39]
	Total Δ133p53 mRNAs expression is correlated with higher risks of recurrence	[40]
	Elevation of total Δ133p53 mRNAs is associated with shorter disease-free survival	[41]
Cholangiocarcinoma	Total Δ133p53 mRNAs upregulation, and decreased TA mRNAs, is associated with shortened overall survival	[42]
	Total Δ133p53 mRNAs reduction sensitizes to 5-Fluorouracil treatment	[43]
Lung carcinoma	Total Δ133p53 mRNAs are overexpressed in tumor as compared to adjacent non-cancerous tissue	[44]
Esophageal Squamous Cell Carcinoma	High tissue or serum total Δ133p53/TAp53 ratio is associated with poor overall and progression-free survival.	[45]
Ovarian cancer	Total Δ133p53 mRNAs expression in mutant *TP53* patients is associated with longer overall survival and disease-free survival	[46]
	Total Δ133p53 mRNAs are associated with increased overall survival (and borderline disease-free survival) independently of *TP53* mutation status	[47]
Renal cell carcinoma	Wild-type tumors correlate with worse overall survival. Total Δ133p53 mRNAs are downregulated in WT tumors compared to mutant tumors and normal adjacent tissue	[48]
**Cancer Type**	**Single Specific Δ133p53 Isoform Association with Disease**	**Ref.**
Breast cancer	Δ133p53α mRNA is detected in tumor but not in normal tissue	[6]
	Δ133p53β mRNA detection is associated with worse overall and disease-free survival	[49]
	Δ133p53β protein is overexpressed in invasive tumor as compared to non-invasive ones	[50]
Glioblastoma	Δ133p53β is elevated in glioblastoma malignant cells and is associated with immunosuppressive and chemoresistant environment	[36]
Prostate cancer	Δ133p53β mRNA is elevated in tumor vs. non-neoplastic tissue and is associated with shorter progression-free survival	[37]
Melanoma	Increased Δ133p53β mRNA expression is associated with poorer overall survival	[51]

The first part of the table displays the studies in which all three Δ133p53 mRNAs have been quantified as a whole, while the second part of the table displays the studies in which a single specific isoform has been identified.

**Table 2 cancers-12-03422-t002:** Δ133p53α is non-mutagenic and non-oncogenic in normal human cells.

Δ133p53α is	Because Δ133p53α	Ref(s)
Non-mutagenic	Enhances DNA double-strand break repair	[37,42,51,52,59,60]
	Does not induce chromosomal and microsatellite repeats abnormalities	[52]
	Does not cause increased mutation rate, unlike canonical p53α knockdown	[52]
Non-oncogenic	Does not prevent p53α-dependent apoptosis of severely damaged cells	[52]
	Increases replicative lifespan without immortalizing cells	[23,24,25,37,50,51,52]
	Does not induce malignant transformation	[23,24,25,37,50,51,52]

**Table 3 cancers-12-03422-t003:** Biological activities regulated by Δ133p53α, Δ133p53β and Δ133p53γ.

Biological Function	Δ133p53α	Δ133p53β	Δ133p53γ
Regulation	Ref(s)	Regulation	Ref(s)	Regulation	Ref(s)
Proliferation	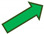	[6,15,23,24,25,39,53,54,61,62,63,64]	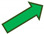	[36,40,41,65]	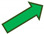	[41,49,66]
Normal stem cells pluripotency	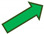	[53,54,64]	ND	-	ND	-
Cancer stemness	ND	-	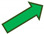	[65]	ND	-
Cellular senescence	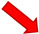	[23,24,25,39,54,61,62]	ND	-	ND	-
DNA repair	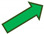	[39,52,54,60,62,67]	ND	-	ND	-
Angiogenesis	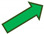	[66]	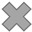	[66]	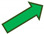	[66]
Motility/Invasion	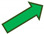	[41,49,66,68]	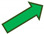	[41,49]	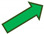	[41,49,66]
Immune cells infiltration	ND	-	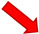	[36,37]	ND	-

Green upward arrows indicate induction, while red downward arrows stand for downregulation of the biological function. ND; Not Determined. The grey cross means that the isoform has been shown to be uninvolved in the biological function.

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
