# Peer review of "The Δ133p53 Isoforms, Tuners of the p53 Pathway"

_cancers, 2020, doi:10.3390/cancers12113422_

Round 1
Reviewer 1 Report
The review is a comprehensive study, well organized, and clear. It covers all different perspectives related to the D133p53 isoforms. I think it would be useful to update the bibliography by adding recent papers about these isoforms. As an example:
1) "p53 isoform Δ113p53 promotes zebrafish heart regeneration by maintaining redox homeostasis" by Ye et al., 2020 (probably in the zebrafish paragraph)
2) "Expression of p53 N-terminal isoforms in B-cell precursor acute lymphoblastic leukemia and its correlation with clinicopathological profiles" by Oh et al., 2020
3) "Splicing factor DHX15 affects tp53 and mdm2 expression via alternate splicing and promoter usage" by Mc Elderly et al., 2019
4) "Induction of p73, Δ133p53, Δ160p53, pAKT lead to neuroprotection via DNA repair by 5-LOX inhibition" by Shekhar and Dey, 2020
5) "Functional interplay between p53 and Δ133p53 in adaptive stress response" by Gong et al., 2020
Author Response
Dear reviewer,
Thank you for your suggestions.
The articles "Splicing factor DHX15 affects tp53 and mdm2 expression via alternate splicing and promoter usage" by Mc Elderly et al., 2019 and "p53 isoform Δ113p53 promotes zebrafish heart regeneration by maintaining redox homeostasis" by Ye et al., 2020 have been referenced as ref. [89] and [90], respectively, and discussed in the zebrafish section which now reads:
“Furthermore, in zebrafish, Def gene mutations selectively up-regulates Δ113p53 expression, leading to impaired development of digestive organs[80]. A similar mechanism has been described with mutation of DHX15, a splicing factor, which up-regulates Δ113p53 expression and leads to morphological defects and embryo lethality[89]. On the contrary, in differentiated tissues, homeostasis is maintained by the pluripotency and differentiation potential of resident cells, which is favored by Δ133p53α/Δ113p53. Importantly, a recent report demonstrated that Δ113p53 promotes zebrafish heart regeneration[90]. It accumulates in cardiomyocytes at the injury site, favoring their proliferation and the maintenance of redox homeostasis, thus promoting myocardial regeneration.”
The article "Expression of p53 N-terminal isoforms in B-cell precursor acute lymphoblastic leukemia and its correlation with clinicopathological profiles" by Oh et al., 2020 has been added as ref. [47] in the Δ133p53 isoforms in cancers section which now reads:
“Similar upregulation of total Δ133p53 mRNAs has been observed in B-cell precursor acute lymphoblastic leukemia[47] and in lung carcinomas as compared to adjacent non-cancerous tissue[48] and is reported to increase cancer cell survival in pathogen-driven cancers such as gastric tumors associated with H. pylori[49].”
The article "Induction of p73, Δ133p53, Δ160p53, pAKT lead to neuroprotection via DNA repair by 5-LOX inhibition" by Shekhar and Dey, 2020 was already referenced in the manuscript in the section Biological activities / Δ133p53α / Δ133p53α in DNA repair. It appears as ref. [69] in the revised manuscript (page 8, line 272).
Nevertheless, at this time we hesitate to include the article "Functional interplay between p53 and Δ133p53 in adaptive stress response" by Gong et al., 2020, which reports a newly discovered mouse isoform named Δ123p53. This report includes many important findings and has important implications for our planned mouse experiments and future publications. However, it is not prioritized in our review article that focuses on endogenous human isoforms. Future studies will clarify the implications of this mouse isoform on human isoform studies. We will for sure refer to this paper wherever more relevant in future.
Reviewer 2 Report
In the article entitled: “The Δ133p53 isoforms, tuners of the p53 pathway” by the authors Sebastien M. Joruiz, Jessica A. Beck, Izumi Horikawa and Curtis C. Harris, the authors give an overview of current knowledge on Δ133p53 isoforms (encompassing Δ133p53α, Δ133p53β and Δ133p53γ). The review is very nicely and precisely written, giving an introduction of the structure, following by regulation of gene and protein expression, and detailed overview of current detection methods. Their role in cancer is comprehensively explained for each isoform individually and mutually compared. Furthermore, the evolutionary preservation is described in drosophila, mouse, and zebrafish. Finally, in Conclusion and perspectives chapter, the authors propose a model for the functions of all Δ133p53 isoforms where the summarized everything that is known so far about these N-terminally truncated isoforms.
The review is absolutely acceptable for publication.
However, there are few minor points to correct:
Line 51, full stop is missing
Line 64, C-terminal (not c-terminal)
Line 74, “This results in reduced production of the ____and γ isoforms.” (β is missing)
Table 1. Ref. - in one line
Table 3. Regulation - - in one line
Line 464. Insert a space between end of legend and text body. And, please, tab the Figure 3. legend to distinguish it!
Author Response
Dear reviewer,
Thank you for pointing this out. Each of the changes suggested have been corrected.
Reviewer 3 Report
This represents a nice review of the current understanding of this p53 isoform written by experts in the field.
Author Response
Dear reviewer,
Thank you for your time and your kind comments
Reviewer 4 Report
The authors present and excellent review on a very interesting topic for everyone working in the p53 field, as well as the general cancer area of research. p53 isoforms are still a poorly explored topic, but they have a strong impact on cancer. Thus, a thorough review on the delta133 isoforms was long overdue.
The review is exhaustive and clearly organized. The figures and tables are relevant and well produced. The only minor revision I would suggest would be to discuss a bit more the mechanism by which these isoforms would perform their functions. Most of it is clearly unknown, but even a comment in this direction would be useful. For instance, the fact that an isoform that has no transactivation domains can induce gene expression independently of p53 is puzzling. Also, how can the isoforms be all ubiquitylated (page 3, line 135) but not degrades (page 3, line 114). Some discussion and hypothesis in these areas could be very interesting.
Author Response
Dear reviewer,
Thank you for your suggestions.
To address your question about how the isoforms can all be ubiquitylated (page 3, line 144 of revised version) but not degraded (page 3, line 123), we modified the text, and discussed the hypothesis that such ubiquitylation may be linked to proteasome-independent functions. In support of this hypothesis, we referenced “Proteasome-independent functions of ubiquitin in endocytosis and signaling.” Science 2007 by Mukhopadhyay et al., and “Non-traditional functions of ubiquitin and ubiquitin-binding proteins.” J Biol Chem 2003 by Schnell et al.
The revised text reads as follows (modifications are in red)
“In addition, as mentioned earlier, Camus et al. have reported that all p53 isoforms are ubiquitylated without necessarily promoting their degradation[20]. This suggests that ubiquitylation of p53 isoforms may also be associated with proteasome-independent functions, including the regulation of subcellular location and protein interaction[33,34], further underscoring the potential for PTMs to modulate the stability and activity of p53 isoforms.”
To discuss the mechanism by which the Δ133p53 isoforms would perform their functions independently of canonical p53, we hinted in the previous version of the manuscript that this could be through interaction with p73 isoforms: “Following genotoxic stress, Δ133p53α accumulates[8,11] and leads to upregulation of repair genes, such as LIG4, RAD51, and RAD52 and the activation of DNA repair pathways[44,66]. This has been suggested to represent a p53α-independent function of Δ133p53α that occurs through interaction with the p53 family member p73[66].” (page 8, line 266).
Nevertheless, we agree that this was not enough to address this important question. Therefore, we added some more discussion and hypothesis.
In the section Biological activities / Δ133p53α / Δ133p53α in cancer biology (page 8, line 296), we hypothesize that the p53α-independent activity of the Δ133p53 isoforms may be carried through interaction with other p53 family members. In support of this hypothesis, we referenced “Differential effects of diverse p53 isoforms on TAp73 transcriptional activity and Apoptosis.” Carcinogenesis, 2013 by Zorić et al. The revised text now reads:
“In the context of cancer, Δ133p53α and Δ133p53γ, but not Δ133p53β, can promote angiogenesis and subsequent metastasis by up-regulating expression of pro-angiogenic genes and repressing anti-angiogenic genes[72]. This is another critical function of Δ133p53α which has been suggested to occur independently of p53α although the exact mechanism is unknown. Since the Δ133p53 isoforms lack transactivation domains, a possible explanation for this activity could be though interaction with p53 family members other than p53α. For example, it was shown that all three Δ133p53 isoforms can interact with TAp73α and TAp73β and counteract TAp73β transactivation function in a promoter-dependent manner[73]. Interestingly, Δ133p53α and Δ133p53β inhibits TAp73β transactivation of pro-apoptotic genes[73].”
Furthermore, in the conclusion remarks of the Biological activities section, we added the hypothesis that the specific DNA-binding ability of the Δ133p53 isoforms may also have a role: (page 9, line 338)
“The specific DNA-binding ability of Δ133p53α[44] and Δ133p53β[18] may contribute to some of the common and unique functions.”